# Near Infrared Spectroscopy Detects Change of Tissue Hemoglobin and Water Levelsin Kawasaki Disease and Coronary Artery Lesions

**DOI:** 10.3390/children9030299

**Published:** 2022-02-22

**Authors:** Ho-Chang Kuo, Shih-Feng Liu, Pin-Xing Lin, Kuender D. Yang, Bor-Shyh Lin

**Affiliations:** 1Kawasaki Disease Center and Department of Pediatrics, Kaohsiung Chang Gung Memorial Hospital, Kaohsiung 83301, Taiwan; erickuo48@yahoo.com.tw (H.-C.K.); liuphysico@yahoo.com.tw (S.-F.L.); 2College of Medicine, Chang Gung University, Taoyuan 33302, Taiwan; 3Department of Respiratory Therapy, Kaohsiung Chang Gung Memorial Hospital and Chang Gung University College of Medicine, Kaohsiung 83301, Taiwan; 4Institute of Imaging and Biomedical Photonics, National Yang Ming Chiao Tung University, Tainan 711, Taiwan; a12321d@gmail.com; 5Department of Medical Research, MacKay Children’s Hospital, Taipei 104, Taiwan; yangkd.yeh@gmail.com; 6Department of Microbiology & Immunology, National Defense Medical Center, Taipei 114, Taiwan; 7Institute of Clinical Medicine, National Yang Ming University, Taipei 114, Taiwan

**Keywords:** Kawasaki disease, infrared, spectroscopy

## Abstract

Background: Kawasaki disease (KD) is a form of systemic vasculitis that mainly affects children under the age of five years old. Limb swelling and redness are among the primary symptoms of KD. Previous studies have reported that wireless optical monitoring systems can identify limb indurations characteristics in patients with KD. Therefore, we conducted this study to monitor the dynamic changes in different stages of KD and the disease outcome of coronary artery lesions (CAL).Methods: KD patients who were admitted for intravenous immunoglobulin (IVIG) treatment and controls with or without fever were enrolled in this study. Near infrared spectroscopy data were collected for KD patients at different stages, including before (within one day before IVIG treatment, KD1) and shortly after IVIG treatment (within three days, KD2), at least 21 days after IVIG (KD3), 6 months later (KD4), 1 year later (KD5), 2 years later (KD6), and 3 years later (KD7).Results: This study included a total of 350 pieces of data, including data from 20 healthy controls, 64 fever controls, 53 KD1, 67 KD2, 58 KD3, 28 KD4, 25 KD5, 15 KD6, and 20 KD7. The relative HbO_2_ of the KD1 group were significantly lower than those of the healthy group (0.298 ± 0.01 vs. 0.304 ± 0.05, *p* = 0.028) but no significant differences were found with the fever group. The HbT concentrations of KD1 group showed significantly lower than health group (0.632 ± 0.019 vs. 0.646 ± 0.021, *p* = 0.001) but no significant difference with fever control. Relative levels of HbO_2_, HbT and Hb showed significant difference between KD1 and health control while StO_2_ and H_2_O showed difference between KD1 and fever control. The relative H_2_O concentration was significantly higher in KD patients with CAL formation than without (*p* < 0.005). Conclusion: This report is the first to use near infrared spectroscopy to detect changes in tissue hemoglobin and water levels at different stages of KD in patients and showed that water content was significantly associated with CAL formation. This non-invasive device may benefit physicians by serving for early identification of KD from fever illness.

## 1. Introduction

Kawasaki disease (KD) is a form of acute febrile multisystem vasculitis with an unknown etiology that primarily presents in children under the age of five years old. In recent years, KD has been the leading cause of acquired heart disease in children in developing countries, especially in Asia [1,2]. Previous reports have indicated that KD patients have a lower hemoglobin concentration and an even lower one after treatment with intravenous immunoglobulin (IVIG) [3]. The American Heart Association (AHA) suggested anemia (by age) as a supplementary criterion for suspecting KD when fever persisted for more than five days in children with two or three compatible KD clinical criteria (or more than seven days in infant without other explanation) [4]. Hepcidin-induced iron deficiency and high-dose aspirin were reported to be related with transient anemia in KD [5,6,7]. Liu et al. reported that anemia demonstrated a significant difference in differentiating KD from sepsis in a nomogram prediction model [8]. Anemia or hemoglobin levels play a very important role in distinguishing KD from other fever illness in children.

Induration and redness on the extremities are the primary symptoms of acute-stage KD, and the AHA statement suggested that in the presence of more than four of the principal clinical criteria, particularly when redness and swelling of the hands and feet are present, the diagnosis of KD may be made with only four days of fever [9]. The AHA statement emphasized the importance of limb induration in KD patients and helped to distinguish KD from other fever illness. Currently, objective detection methods for measuring the induration of KD patients’ extremities are still lacking. From our previous report, the relative water (H_2_O) concentrations in the palm tissue of patients with KD showed to be better understand the status of blood circulation and angioedema. The increased relative water concentrations found in KD patients than control subjects are due to patients with KD presenting with angioedema or tissue swelling, which is not a common symptom in other fever illness of children [10].

Near infrared spectroscopy (NIRS) is a non-invasive approach for estimating changes in human tissue composition, such as hemoglobin concentrations, water content, etc., and has been widely used in various biomedical applications [11,12]. Therefore, the NIRS technique can monitor the state of blood circulation and angioedema in the tissue and has the potential to further evaluate the state of KD. From our previous report, lower relative total hemoglobin (HbT) concentrations and greater relative water concentrations were found in the acute stage of KD patients when compared with control subjects [10]. Liu et al. [8] reported that anemic condition alone does not allow an unambiguous KD diagnosis. Relative Hb levels as well as water levels in the tissue detection by NIRS could provide better differentiation for KD from other fever illness. Patients with KD presented with angioedema, which is not a symptom in other fever illnesses. This study was conducted to analyze the dynamic change of different stages of KD and disease outcome.

## 2. Methods

### 2.1. Patients Studied

This cross-sectional study consisted of children who met the KD criteria and were admitted to Kaohsiung Chang Gung Memorial Hospital for treatment. The Institutional Review Board of Chang Gung Memorial Hospital approved this study (No. 103-3133B) on Aug 18th of year 2014 and we obtained informed consent from the parents or guardians of all of the participating children. Data were collected from KD patients in different stages, including KD1 (within one day before IVIG treatment), KD2 (within three days after IVIG), KD3 (at least 21 days after IVIG), KD4 (six months after IVIG), KD5 (one year after IVIG), KD6 (two years after IVIG), and KD7 (three years after IVIG). Any patients whose symptoms did not fit the criteria were excluded. The age matched fever control (FC) group consisted of patients admitted to the hospital with an acute infection, such as acute pharyngitis, acute tonsillitis, croup, acute bronchitis, pyuria, or acute bronchiolitis, while the age matched healthy controls (HC) were those who had no fever or any other active signs of infection. The CAL was defined as a coronary artery with an internal diameter of at least 3 mm (4 mm if the patient was older than five years) or a segment with an internal diameter at least 1.5 times larger than that of an adjacent segment, as identified through echocardiography. All subjects were initially treated with a 12-h period single dose of IVIG (2 g/Kg). Aspirin (3–5 mg/kg/day) was given until all signs of inflammation had resolved or regression of CAL was observed on two-dimensional (2D) echocardiography.

### 2.2. Optical Device for Evaluating the Tissue of Patients and Controls

The basic scheme of the proposed optical device for evaluating the tissue of KD patients and controls is illustrated in Figure 1. It mainly consists of two parts, namely an optical probe and a wireless signal acquisition module. The optical probe contained a multi-wavelength surface mounted device (SMD) light emitting diode (LED) and photodiode (PD) to provide a tri-wavelength (700 nm, 910 nm, and 950 nm) light source and a light detector. The wireless signal acquisition module was designed to drive these LEDs, receive and digitize the reflected light signal, and then transmit the optical signal to the back-end laptop platform wirelessly via Bluetooth. After receiving the optical signal, the back-end laptop platform estimated the hemoglobin concentration and water content in the tissue.

When light penetrates human tissue, the photons may be scattered or absorbed by different human tissue components to result in optical attenuation. The absorbing and scattering abilities of human tissues depend on the wavelength of light and the type of human tissue component. For the near-infrared light, except for hemoglobin and water, most human tissue components provide lower absorbing properties. According to the difference between their absorbing spectra, the change of oxy-hemoglobin (HbO_2_), deoxy-hemoglobin (Hb) concentrations, and water (H_2_O) can then be estimated by a modified Beer-Lambert law [13] based on the optical attenuation of the multi-wavelength near-infrared light. After estimating the HbO_2_ and Hb concentrations, the total hemoglobin (HbT) concentration related to the blood volume in the tissue can be defined as the sum of HbO_2_ and Hb concentrations, while tissue oxygen saturation ($S_{t}O_{2}$) is defined as the proportion of the oxyhemoglobin to the total hemoglobin.

In this study, the proposed wireless optical device was placed on the flexor pollicis brevis to measure the changes of relative HbO_2_, Hb, HbT, and H_2_O concentrations and StO2 in the tissue. The measuring depth was about 5 mm. The area under the receiver operating characteristic (ROC) curve was used to predict CAL formation. We adopted the nonparametric statistics, t-test and analysis of variance (ANOVA) method to analyze significant difference (*p* < 0.05).Data were shown as mean with standard deviation (SD) using SPSS 12.0 software to perform analysis.

## 3. Results

### 3.1. Changes of Physiological Parameters in Different Stages of KD

The demographic data of all participants are shown in Table 1. The changes of relative HbO_2_, Hb, HbT, StO_2_, and H_2_O concentrations in different disease stages of KD are showed in Figure 2a–e, respectively. The relative HbO_2_ of the KD1 group were significantly lower than those of the healthy group (0.298 ± 0.01 vs. 0.304 ± 0.05, *p* = 0.028), but no significant differences were found with fever group (0.295 ± 0.013, *p* = 0.26). The HbT concentrations of KD1 group were significantly lower than in the health group (0.632 ± 0.019 vs. 0.646 ± 0.021, *p* = 0.001) but no significant differences were found with the fever control group (0.633 ± 0.016, *p* = 0.77). After the acute stage of KD1, both the relative HbO_2_ and HbT concentrations gradually increased to similar values as the healthy group. The relative Hb concentration of the KD1 group was significantly lower than that of the healthy group (0.334 ± 0.011 vs. 0.342 ± 0.009; *p* = 0.006), and were still low in KD2, but increased significantly after the KD3 to the KD7 stage (KD 1 vs. KD 4, *p* = 0.007). The difference of StO_2_ showed significant lower in fever group (46.678 ± 1.359, KD1 vs. FC, *p* = 0.009) when compared with the KD1 group (47.246 ± 1.268) and healthy group (47.145 ± 1.047, KD1 vs. HC, *p* = 0.74). The relative H_2_O concentration of the KD1 group (75.138 ± 3.937) was significantly higher than that of fever group (73.638 ± 3.508; KD1 vs. FC, *p* = 0.019) but no significant differences were found with the healthy group (75.560 ± 2.509; KD1 vs. HC, *p* = 0.640).The same group of children were longitudinally followed over time with KD1 and KD3 (N = 12) and showed significant association in HbT (*p* = 0.015, paired-sample *t* test) but not in others. Relative levels of HbO_2_, HbT, and Hb showed significant differences between KD1 and the health control group, while StO_2_ and H_2_O showed differences between KD1 and fever control. The ANOVA analysis in HbO_2_ (*p* < 0.0001), Hb (*p* < 0.0001), HbT (*p* < 0.0001), StO_2_ (*p* < 0.0001), and H_2_O (*p* = 0.031) showed significant difference between groups of HC, FC, and KD1-KD7.

### 3.2. Relationship between Coronary Artery Lesions and Water Concentration

The relative water concentration was significantly higher in KD with CAL formation (N = 9) than without (N = 24) (*p* < 0.005, Figure 3) in the KD1 stage. The difference between KD1 and KD2 relative H_2_O concentration levels also demonstrated significant differences with regard to CAL formation (*p* = 0.03). The significance which emerged for the |KD1-KD2| H_2_O concentration between groups with and without CAL maybe a reflection of the significant difference observed for KD1 patients. The area under the receiver operating characteristic (ROC) curve was used (relative H_2_O: 75.76) to predict CAL formation and showed significant association (20.5% vs. 48.3%, *p* = 0.016).The relative KD1 levels of HbO_2_ (*p* = 0.002) and HbT (*p* = 0.011) was also significantly higher in KD patients with CAL formation than without CAL, but was not found in other stage of KD. The relative H_2_O concentration levels of limbs showed a positive association with GPT levels (Glutamic-Pyruvic Transaminase or ALT, alanine aminotransferase, *p* = 0.026), while Hb, HbO_2_, and HbT were all found to have a positive association with serum levels of albumin (*p* < 0.01). After multiple logistic regression analysis (including HbO_2_, Hb, HbT, SPO_2_, H_2_O, CRP, albumin, total white blood cell count, GOT and GPT), HbO_2_ was determined to be the independent risk factor for CAL formation in KD patients. The CAL formation showed regression in 78% (7/9) of patients after two months of follow up. Male gender has a higher risk for CAL formation (*p* = 0.047) when compared with female gender KD patients. There were no significant differences in regarding age distribution in CAL formation of KD (*p* = 0.81). The power analysis showed 0.92 in HbO_2_ (*p* = 0.001) and 0.79 in HbT (*p* = 0.004) in KD1 to predict CAL formation.

## 4. Discussion

KD has become the leading cause of acquired heart disease in children among developing countries. Early detection of potential KD and predicting the disease outcome of coronary artery involvement are the most important issues with regard to this problem. In a previous report, we showed decreased relative HbT levels and increased water concentrations in patients with KD when compared with fever illness. Lower relative HbT concentrations in patients with KD suggest anemia in patients with KD and are related to inflammation status. Induration of limbs (hands and feet), including swelling and redness, is an important symptom of KD. Currently, no objective device is available for detecting the status of limbs induration in KD. Although the relative HbT levels did not demonstrate an association with Hb levels in peripheral blood examination, it did show a significant association with albumin levels that was a powerful predictor for high-risk KD [14].

The relative H_2_O concentration level of limbs was found to be significantly higher in KD patients than in controls and in the KD groups with CAL formation. It also revealed a significant positive association with GPT levels. Chen et al. reported that sonographic gallbladder abnormalities were associated with higher C-reactive protein, GPT, neutrophil, and IVIG resistance in KD [15]. Liu et al. reported that alanine transaminase (GPT) was a major factor in the nomogram model and had high precision for distinguishing KD from fever controls [16]. The aminotransferase (AST)/ALT ratio and ALT levels in KD were reported to be significantly associated with disease severity, IVIG resistance, and CAL formation [17,18]. The age or gender difference between the different stage group of KD and control groups may have impact on the levels ofH_2_O, HbO_2_, HbT, StO2, and Hb, and for this reason we enrolled age-matched with KD1 and analyzed the gender difference between variables (there were no significant difference in variables between male and female gender, only higher H_2_O in KD2 male than female gender, *p* = 0.039).

The wireless near-infrared spectroscope (NIRS) was reported to be used for assessing Buerger’s exercise on dorsal foot skin circulation in patients with vasculopathic diabetic foot ulcers [19], cerebral ischemia, and cerebral perfusion pressure in an animal model of traumatic brain injury [20], and a cardiac arrest model for muscle oxygenation, mental stress levels, etc. [21]. In this study, we first monitored limb induration by using wireless NIRS in different stages of KD from the acute stage prior to IVIG treatment to the chronic stage of more than three years, which was followed up with in order to reveal the dynamic changes of relative HbO_2_, HbT, SatO_2_, and H_2_O. The H_2_O concentration was found to be significantly higher in KD1 (before IVIG treatment) than controls and even higher in those KD patients with CAL formation. From previous reports, serum albumin levels and C-reactive protein/albumin ratio showed significant association with CAL formation and/or IVIG resistance in KD patients [14,22]. Increased microvascular permeability, vascular leakage, and vascular leakage caused hypoalbuminemia and represent important pathogeneses of noncardiogenic edema in KD [23]. The limitation of this study is case number of KD patients with CAL formation. Since dryness of skin differs between children of different ages and within the same group of children, patients should be longitudinally followed up with over time, and more cases are needed to make further conclusions regarding this problem.

The NIRS monitoring system may provide an easier and more convenient way to determine KD in children with fever and predict coronary artery lesions earlier for more aggressive treatment in order to prevent life-long complications.

## 5. Conclusions

This report is the first to show dynamic changes of tissue content, with respect to water concentration and relative hemoglobin levels in limbs of KD patients, and their association with coronary artery involvement. The wireless near-infrared spectroscope (NIRS) monitoring system may provide a non-invasive method to determine KD in children with fever.

## Figures and Tables

**Figure 1 children-09-00299-f001:**
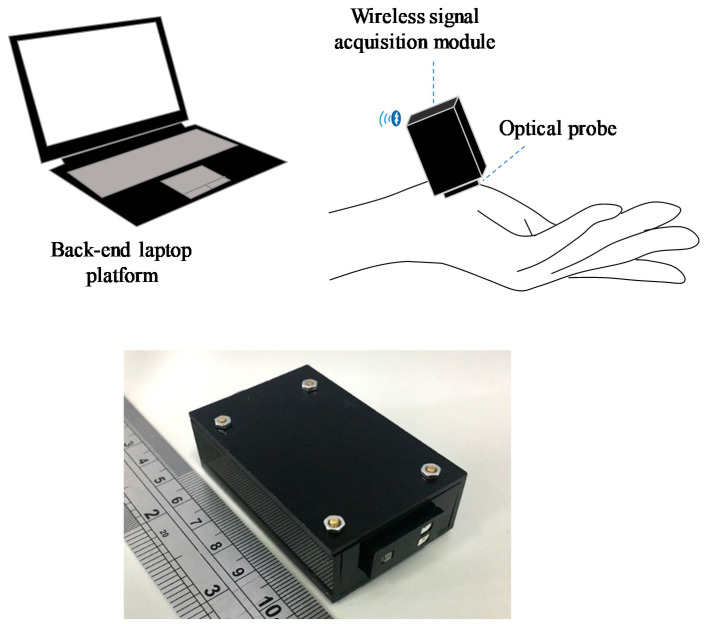
Basic scheme of optical device.

**Figure 2 children-09-00299-f002:**
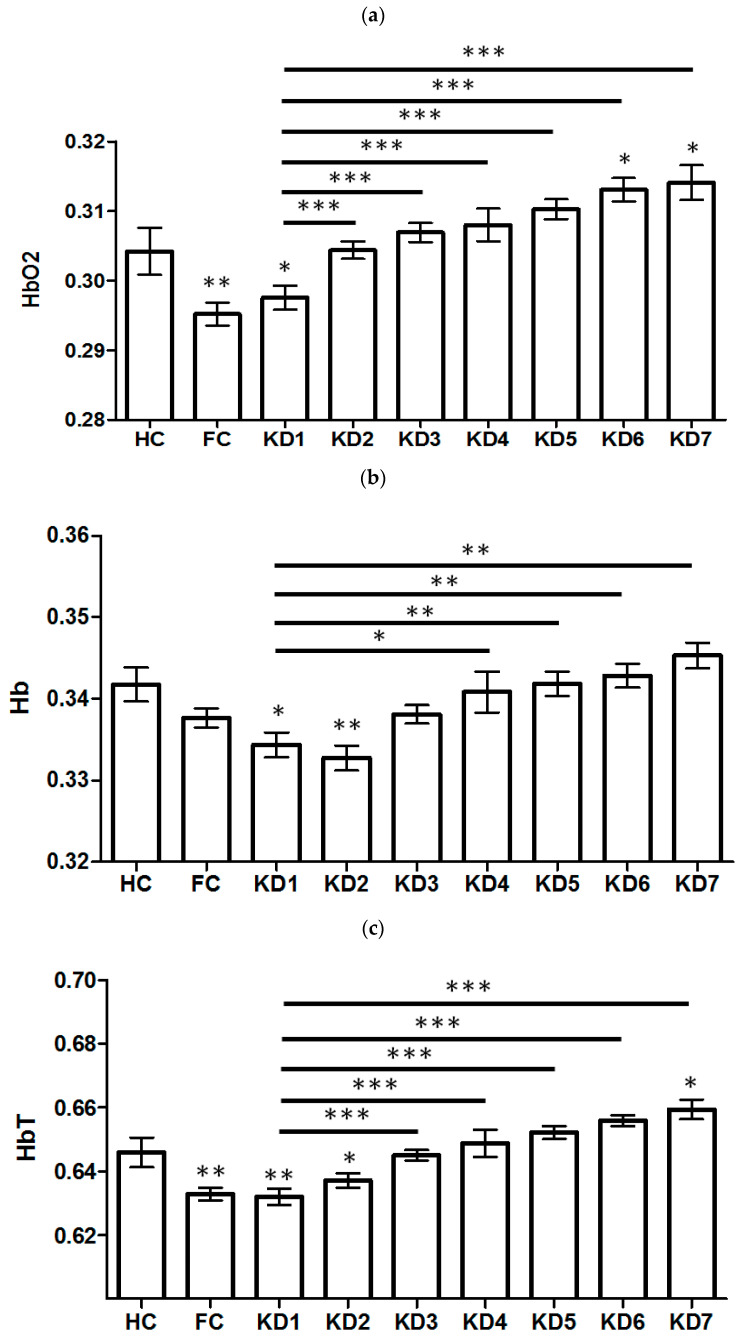
The changes of relative HbO_2_, Hb, HbT, StO_2_, and H_2_O concentrations indifferent disease stages of Kawasaki disease (KD), healthy control (HC), and fever control (FC) are shown in (**a**–**e**), respectively. The relative HbO_2_ of the KD1 group were significantly lower than those of the healthy group. The HbT concentrations of KD1 group were significantly lower than the health group (*p* = 0.001) but not the fever control (*p* = 0.77). After the acute stage of KD1, both the relative HbO_2_ and HbT concentrations gradually increased to similar values as the healthy group. The relative Hb concentration of the KD1 group was significantly lower than that of the healthy group (*p* = 0.006) and were still low in KD2 but increased significantly after the KD3 to the KD7 stage. The difference of StO_2_ showed significant lower in fever group when compared with the KD1 group and healthy group. The relative H_2_O concentration of the KD1 group was significantly higher than that of fever group (*p* = 0.019) but no significant difference with the healthy group. Relative levels of HbO_2_, HbT, and Hb showed significant difference between KD1 and health control while StO_2_ and H_2_O showed difference between KD1 and fever control.KD1, within one day before IVIG treatment; KD2, after IVIG treatment (within three days); KD3, at least 21 days after IVIG; KD4,six months after IVIG; KD5, one year after IVIG; KD6,two years after IVIG; KD7,three years after IVIG; intravenous immunoglobulin (IVIG), * *p* < 0.05, ** *p* < 0.01, *** *p* < 0.001.

**Figure 3 children-09-00299-f003:**
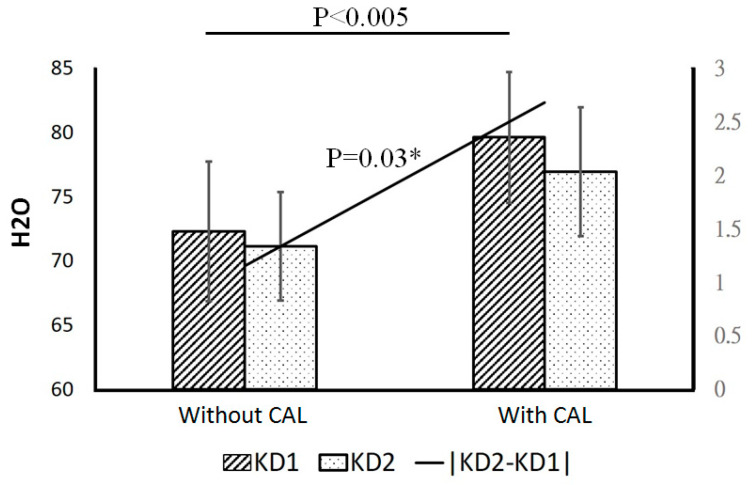
Relationship between coronary artery lesions (CAL) and relative H2O concentration in Kawasaki disease (KD). The relative water concentration was significantly higher in KD with CAL formation (N = 9) than without (N = 24) (*p* < 0.005) in the KD1 stage. The difference between KD1 and KD2 relative H_2_O concentration levels also demonstrated significant differences with regard to CAL formation (*p* = 0.03). (The left *y*-axis was for the parameters corresponding to different groups, the right *y*-axis was for the difference between KD1 and KD2). KD1, within one day before IVIG treatment; KD2, after IVIG treatment (within three days). * *p* < 0.05.

**Table 1 children-09-00299-t001:** Basic information in different groups.

	Groups								
	HC	FC	KD1	KD2	KD3	KD4	KD5	KD6	KD7
Case number	20	64	58	89	115	66	51	25	23
Age (years)	2.17 ± 1.45	1.92 ± 0.99	1.77 ± 1.22	1.99 ± 1.35	1.97 ± 1.57	2.37 ± 1.55	3.24 ± 1.59	4.12 ± 1.11	5.23 ± 2.18
Male gender	15	36	38	63	67	46	35	16	13
Female gender	5	28	20	26	48	20	16	9	10

KD, Kawasaki disease; IVIG, intravenous immunoglobulin; KD1, within one day before IVIG treatment; KD2, after IVIG treatment (within three days); KD3, at least 21 days after IVIG; KD4,six months after IVIG; KD5,one year after IVIG; KD6,two years after IVIG;KD7,three years after IVIG;HC, healthy control; FC, fever control.

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
