# Peer review of "Near Infrared Spectroscopy Detects Change of Tissue Hemoglobin and Water Levelsin Kawasaki Disease and Coronary Artery Lesions"

_children, 2022, doi:10.3390/children9030299_

Round 1

Reviewer 1 Report

The main research results, their theoretical and practical significance, the validity of the conclusions: The results obtained indicate the effectiveness of the diagnostic technique, can be used not only in scientific terms, but also in practical health care. The conclusions of the work are substantiated.
Adequacy of the use of literary sources: Literary sources are used adequately
The quality of the article design (compliance with the editorial requirements, style, consistency, internal unity, language, etc.): The article is well-prepared, the presentation style is good. There is a clear logical sequence of presentation of their own results.

Author Response

Thanks for your comments.

Reviewer 2 Report

This paper shows that measuring relative HbO2 using near-infrared spectroscopy in patients with Kawasaki disease may identify patients with coronary artery disease, but there seem to be some problems with it.

#1 The KD1 group was not significantly different from the fever group in terms of relative HgO2 levels, and relative HgO2 seems to be an indicator of the degree of acute inflammation. If this is the case, how much better is it than markers of inflammation such as CRP or markers such as transaminase as mentioned in the discussion? Please show such data and add it to the patient background as well. Also, it would be easier to understand if you show the clinical background of patients with and without coronary artery lesions. If there is a possibility that coronary artery disease can be identified, please provide a cut-off value to suggest this.

#2 In this paper, is the notation of the data mean±SD or mean±SE? Please make sure to indicate the statistical method.

#3 It seems difficult to understand the statistical significance in Figures 2 and 3. Please illustrate more clearly where the significant differences are.

#4 Are there any limitations in this paper?

#5 Please mention in the text the acquisition of informed consent and the approval of the ethics committee in this study.

Author Response

response to Reviewer 2:

This paper shows that measuring relative HbO2 using near-infrared spectroscopy in patients with Kawasaki disease may identify patients with coronary artery disease, but there seem to be some problems with it.

#1 The KD1 group was not significantly different from the fever group in terms of relative HgO2 levels, and relative HgO2 seems to be an indicator of the degree of acute inflammation. If this is the case, how much better is it than markers of inflammation such as CRP or markers such as transaminase as mentioned in the discussion? Please show such data and add it to the patient background as well.

  • We have added analysis in the result section.

   Also, it would be easier to understand if you show the clinical background of patients with and without coronary artery lesions.

  • We have added the clinical background of KD patients with and without CAL in the result section.

If there is a possibility that coronary artery disease can be identified, please provide a cut-off value to suggest this.

àWe have added the cut-off value by ROC and showed in the result section.

#2 In this paper, is the notation of the data mean±SD or mean±SE? Please make sure to indicate the statistical method.

  • We have showed in the method section (mean±SD).

#3 It seems difficult to understand the statistical significance in Figures 2 and 3. Please illustrate more clearly where the significant differences are.

  • We have revised the figure 2 and 3.

#4 Are there any limitations in this paper?

  • We have mentioned the limitation in the discussion. (The limitation of this study is case number of KD patients with CAL formation and dryness of skin differs between children and age, more cases are needed to make a conclusion in the further.)

#5 Please mention in the text the acquisition of informed consent and the approval of the ethics committee in this study.

  • We have added in the text.

Reviewer 3 Report

  1. A statistic test for multiple comparisons should be applied, due to the multiple pairwise t-tests performed. It is possible that these findings are due to type I error
  2. Are the participants in each KD group the same children measured at different time points up to three years after the diagnosis? If not, then a power analysis should be performed; it is unlikely that reliable results can result from this study design. Ideally, the same group of children should be longitudinally followed over time.
  3. No data on how coronary lesions were diagnosed are given in the Methods section or how these children were treated, as well as their outcome.
  4. The introduction presents in detail the rationale of COVID-19 infection provoking Kawasaki disease like symptoms, but there are no data about the percentage of children that actually have this pathology, their characteristics, treatment, outcomes and whether this group was comparable to non COVID-19 patients with Kawasaki disease

Author Response

Reviewer 3:

ï‚·  A statistic test for multiple comparisons should be applied, due to the multiple pairwise t-tests performed. It is possible that these findings are due to type I error. ï‚·  Are the participants in each KD group the same children measured at different time points up to three years after the diagnosis? If not, then a power analysis should be performed; it is unlikely that reliable results can result from this study design. Ideally, the same group of children should be longitudinally followed over time.

àWe have added pair-sample t test in the result section.

ï‚·  No data on how coronary lesions were diagnosed are given in the Methods section or how these children were treated, as well as their outcome.

  • We have added the definition of CAL and treatment in the method section and regression of CAL in result section.

ï‚·  The introduction presents in detail the rationale of COVID-19 infection provoking Kawasaki disease like symptoms, but there are no data about the percentage of children that actually have this pathology, their characteristics, treatment, outcomes and whether this group was comparable to non COVID-19 patients with Kawasaki disease.

  • We have revised the introduction section.

Round 2

Reviewer 2 Report

The points I pointed out in the previous article have been properly corrected. One thing, if you have done ROC analysis, please mention it in the statistics section.

Author Response

We have added the ROC analysis it in the statistics section.

Reviewer 3 Report

The authors have not adequately responded all the points made during the initial review round:

  1. A test for multiple comparisons has not been made
  2. Power analysis has not been performed
  3. This point is adequately addressed
  4. The paragraph about the impact of COVID-19 should be removed, as no data about the incidence of this infection are available. What is discussed is therefore speculative.

Author Response

1.A test for multiple comparisons has not been made

--> we have added ANOVA analysis.

2. Power analysis has not been performed

--> we have added the power analysis.

-->We have added in the limitation " the same group of children should be longitudinally followed over time"

3. This point is adequately addressed

4. The paragraph about the impact of COVID-19 should be removed, as no data about the incidence of this infection are available. What is discussed is therefore speculative.

--> we have removed the paragraph of COVID-19.